# A maximum of two readily releasable vesicles per docking site at a cerebellar single active zone synapse

Melissa Silva, Van Tran, Alain Marty*

Université Paris Cité, SPPIN-Saints Pères Paris Institute for the Neurosciences, CNRS, Paris, France

*For correspondence: alain.marty@parisdescartes.fr

Competing interest: The authors declare that no competing interests exist.

**Abstract** Recent research suggests that in central mammalian synapses, active zones contain several docking sites acting in parallel. Before release, one or several synaptic vesicles (SVs) are thought to bind to each docking site, forming the readily releasable pool (RRP). Determining the RRP size per docking site has important implications for short-term synaptic plasticity. Here, using mouse cerebellar slices, we take advantage of recently developed methods to count the number of released SVs at single glutamatergic synapses in response to trains of action potentials (APs). In each recording, the number of docking sites was determined by fitting with a binomial model the number of released SVs in response to individual APs. After normalization with respect to the number of docking sites, the summed number of released SVs following a train of APs was used to estimate of the RRP size per docking site. To improve this estimate, various steps were taken to maximize the release probability of docked SVs, the occupancy of docking sites, as well as the extent of synaptic depression. Under these conditions, the RRP size reached a maximum value close to two SVs per docking site. The results indicate that each docking site contains two distinct SV-binding sites that can simultaneously accommodate up to one SV each. They further suggest that under special experimental conditions, as both sites are close to full occupancy, a maximal RRP size of two SVs per docking site can be reached. More generally, the results validate a sequential two-step docking model previously proposed at this preparation.

## eLife assessment

The study used slice physiology and modeling to investigate neurotransmitter release at the cerebellar parallel fiber-to-molecular layer interneuron synapse, revealing that each docking site can accommodate up to two synaptic vesicles simultaneously. The evidence presented is **convincing**. These **important** findings validate a two-step docking model and shed light on the mechanisms underlying short-term synaptic plasticity and strategies for achieving synaptic reliability, which plays a critical role in information processing in the brain.

## Introduction

Before undergoing exocytosis, synaptic vesicles (SVs) interact with a number of specialized proteins in the presynaptic active zone (AZ; *Südhof, 2012*). Functional studies using variance/mean analysis of synaptic currents (*Clements and Silver, 2000*) or direct visualization of SV fusion (*Maschi and Klyachko, 2017*; *Sakamoto et al., 2018*) indicate that within individual AZs, exocytosis occurs at a finite number of structures called docking or release sites. The number of docking sites per AZ has been estimated in a wide range of preparations, with results varying over a range of 1–15 (*Malagon et al., 2016*; *Maschi and Klyachko, 2017*; *Neher and Sakaba, 2008*; *Pulido et al., 2015*; *Sakamoto*

*et al., 2018*). Morphologically, each docking site has been proposed to be associated with a cluster of RIM 1/2 (*Tang et al., 2016*), voltage-gated calcium channels (*Miki et al., 2017*), and/or Unc 13/Munc13 (*Reddy-Alla et al., 2017*; *Sakamoto et al., 2018*). In each case, several clusters of the relevant protein occur inside one AZ.

As its name suggests, the readily releasable pool (RRP) contains SVs that are readily released after a short burst of presynaptic action potentials (APs). It is widely accepted that exhaustion of the RRP is a main mechanism underlying synaptic depression during high-frequency neuronal activity (*Betz, 1970*; *Elmqvist and Quastel, 1965*; *Schneggenburger et al., 1999*; *Thanawala and Regehr, 2013*; review: *Zucker and Regehr, 2002*). While the size of the RRP is likely related to the number of docking sites, the exact relation between the two quantities has remained unclear. This relation ultimately depends on the specific nature of the interactions between SVs and docking sites. Under electron microscopy, filaments with various angles and lengths can be seen linking SVs to the AZ membrane (review: *Silva et al., 2021*). Nonetheless, the number and spatial arrangement of docked SV around individual docking sites is unknown.

Recently developed models assuming sequential steps of SV binding to individual docking sites have provided an explanation for a range of synaptic changes at several types of central synapses (*Doussau et al., 2017*; *Eshra et al., 2021*; *Fukaya et al., 2023*; *Kobbersmed et al., 2020*; *Lin et al., 2022*; *Miki et al., 2016*). The basis for these models came from functional and modeling studies at parallel fiber to molecular layer interneuron (PF–MLI) synapses and at the calyx of Held indicating calcium-dependent docking within 5 ms after an AP stimulation (*Lin et al., 2022*; *Miki et al., 2018*; *Miki et al., 2016*; *Taschenberger et al., 2016*). Likewise, rapid freezing electron microscopy studies in cultured hippocampal neurons show that, following an AP stimulation, SVs move rapidly (within 10 ms) to the AZ membrane in a calcium-dependent process (*Chang et al., 2018*; *Kusick et al., 2020*; *Wu et al., 2022*). These results suggest that the RRP comprises two classes of SVs representing the last two steps of preparation before release. One variant of sequential models assumes that a docking

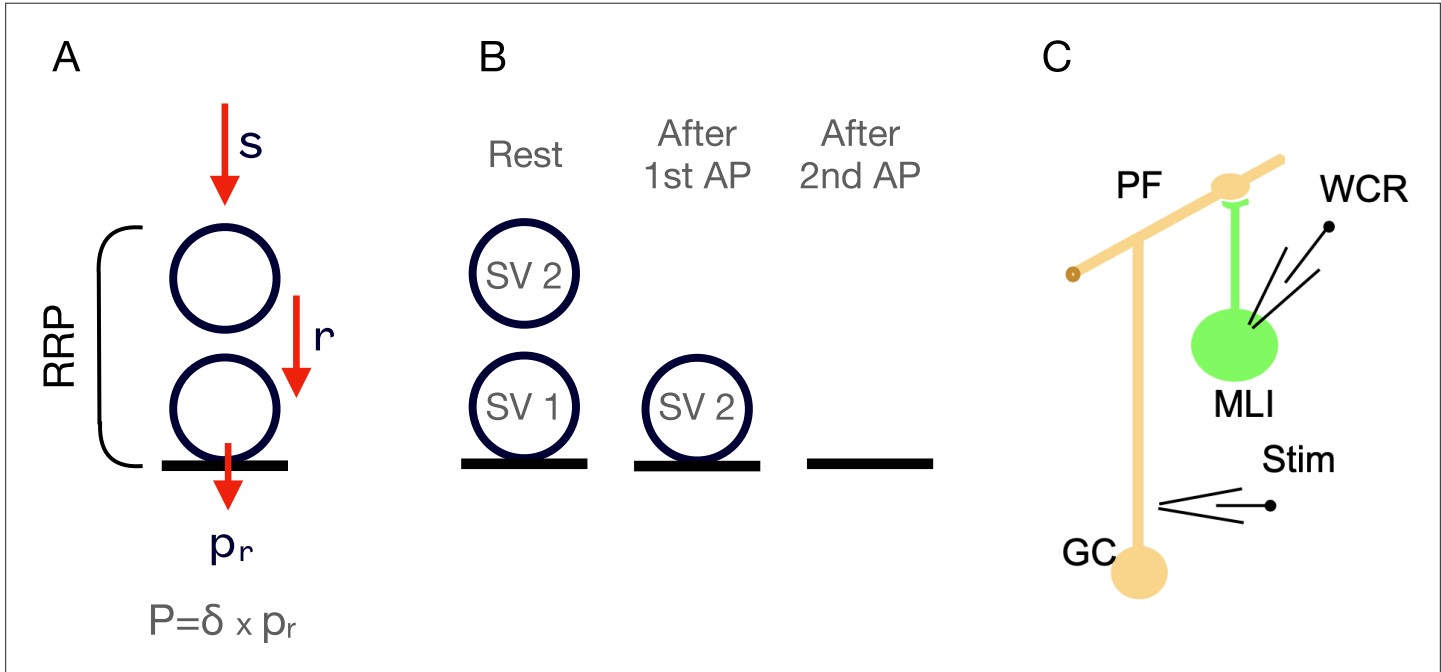

**Figure 1.** Experimental approach to estimate the maximal readily releasable pool (RRP) size at parallel fiber to molecular layer interneuron (PF–MLI) synapses. (**A**) Two-step model of a single release site. At rest, such a site can accommodate up to two synaptic vesicles (SVs): one SV attached to the distal replacement site (RS) and one SV attached to the proximal docking site (DS). The RRP is comprised of the SVs bound to these two sites. Red arrows depict calcium-dependent probabilities of SV movement. SV fusion occurs with a probability of $P$, which depends on $p_r$, the probability of release of a docked SV, and $\delta$, the probability of having an SV bound to the DS (adapted from *Miki et al., 2016*). (**B**) Schematics of RS/DS at rest (left), after one action potential (AP; center), and after two APs (right). After a first AP, the SV that was initially docked (SV1) is released, and the replacement SV (SV2) moves down to the DS (center). After a second AP, the latter SV (SV2) is released (right). (**C**) Experimental arrangement for the recording of simple PF–MLI synapses.

site can accommodate only one SV of either class ('loose state/tight state [LS/TS] model': *Neher and Brose, 2018*), while another variant assumes that a docking site can simultaneously accept two SVs, one of each class ('replacement site/docking site [RS/DS] model': *Miki et al., 2016*).

We recently developed a method allowing us to count SV release events at single PF–MLI synapses reliably and with an excellent time resolution (*Malagon et al., 2016*). In the present work, we take advantage of this to estimate the maximal RRP size in these synapses and to compare it with the number of docking sites. By adjusting experimental conditions, we increase docking site occupancy, release probability, and docking rate. Under these conditions, we find a ratio between RRP size and docking site number close to 2, thus strongly constraining docking site models at this synapse.

## Results

### Theoretical considerations regarding the maximum RRP size per docking site

The purpose of the present work is to measure the maximum RRP size per docking site at PF–MLI synapses and to compare that maximum with the predictions of the RS/DS model (*Miki et al., 2016*).

The basic features of the RS/DS model are pictured in *Figure 1A*. Here, a single docking/replacement unit is shown for simplicity. Incoming SVs first bind to a distal 'replacement site' (upper position in *Figure 1A*), then to an associated proximal 'docking site' (lower position in *Figure 1A*) before being released. The distal replacement site is considered to be ~40 nm away from the plasma membrane, allowing simultaneous occupancy of both distal and proximal docking sites by SVs.

The progression of SVs along the replacement site/docking site assembly during an AP train is driven by the conditional transition probabilities $s$, $r$, and $p_r$ (*Figure 1A*). Each of these probabilities refers to one AP interval during the train. Conditions for an effective transition are: for $s$, that the replacement site is empty; for $r$, that the replacement site is occupied, and that the docking site is empty; for $p_r$, that the docking site is occupied. Thus, if the probability of docking site occupancy before stimulation is $\delta$, the probability that the site releases an SV after the first AP of the train is $P = \delta p_r$ (*Scheuss and Neher, 2001*).

Calcium elevation in presynaptic terminals promotes the replenishment of SVs to emptied docking sites (*Chang et al., 2018*; *Kobbersmed et al., 2020*; *Kusick et al., 2020*; *Neher and Sakaba, 2008*; *Sakaba, 2008*; *Wang and Kaczmarek, 1998*; *Wu et al., 2022*). Accordingly, we assume that calcium increases not only $p_r$, but also $r$ and $s$. As the probability of transition $r$ increases after an AP, SVs can move from the distal to proximal site within a few ms. In the framework of the RS/DS model, due to the high speed of this transition, both replacement and docked SVs are regarded as belonging to the RRP (*Miki et al., 2018*; *Miki et al., 2016*). Likewise, in the related LS/TS model, the RRP comprises the sum of SVs occupying a distal and a proximal SV-binding site, which are called in this model the loosely and tightly docked state (*Lin et al., 2022*; *Neher and Brose, 2018*; *Neher and Taschenberger, 2021*). A key difference between these two models is that the sites in the RS/DS model can be simultaneously occupied while in the LS/TS model a docking site can only accommodate one SV in either state at a time. Thus, the RS/DS model predicts a maximum RRP size per docking site of 2, whereas the LS/TS model predicts a maximum RRP size per docking site of 1.

### Maximizing RRP size and its release during AP trains

We used previously developed methods to record from simple synapses between cerebellar granule cells (GCs) and MLIs (*Figure 1C*). At these synapses, it is possible to count the release of individual SVs with an effective time separation of 0.2 ms between consecutive release events (*Malagon et al., 2016*). It is also possible to count the number of docking sites precisely, giving an opportunity to evaluate the maximum number of released SVs per site.

To determine the maximum number of SVs that a docking site can accommodate, it was essential to maximize the RRP size and its release during an AP train, that is, to maximize $\delta$, $p_r$, and $r$. In the extreme case where $p_r$ and $r$ are both equal to 1, a first AP should release all SVs that were initially in the docking site, and a second AP should release all SVs that were initially in the replacement site (*Figure 1B*; *Tran et al., 2022*). The RRP size is then given by the sum of the number of released SVs for the two APs.

Recordings from simple synapses between cerebellar GCs and MLIs (*Figure 1C*) were initiated in an elevated external calcium concentration (3 mM), a condition that has been shown to elevate $\delta$, $p_r$, and $r$ (*Miki et al., 2018*; *Malagon et al., 2020*). In this condition and for the example shown in *Figure 2A*, responses to the first two stimuli of an 8-AP (8-action potentials) train (100 Hz within each train, 10 s intervals between trains) were mostly comprised between 0 SV (failure) and 2 SVs (*Figure 2A*, left). While keeping 3 mM Ca$_o$, we next performed two modifications aiming at further enhancing release probability and docking site occupancy: we applied the potassium channel blocker 4-amidopyridine (4-AP), which increases $p_r$, and we performed a post-tetanic potentiation (PTP) protocol, keeping 4-AP in the perfusion (see Methods). For the example shown, responses to the first two stimuli were clearly enhanced by 4-AP (now comprised between 2 and 4 SVs, with no failure: *Figure 2A*, center). In contrast, responses to stimuli 6–8 were reduced with respect to the control. We then applied a series of 60 AP trains, each with 8 APs evoked at 100 Hz, with an inter-train interval of 2 s. This protocol (called 'PTP'hereafter) results in an increase in docking site occupancy lasting several minutes (*Tran et al., 2023*). After PTP induction, the responses to the first stimuli increased, or remained similar to those in 4-AP (as in the experiment shown), while responses to later stimuli displayed more synaptic depression. Group results indicate a marked enhancement in the mean released SV numbers after the first two stimuli, and somewhat reduced responses for stimuli 5–8, when comparing results in 4-AP or after PTP with control (*n* = 8; *Figure 2B, C* showing the number of SVs released per AZ and per docking site, respectively).

## Estimating the number of docking sites, *N*, and the release probability, *P*

Following stimulus number *i*, the number of released SVs, $s_i$ (*Figure 2B*), is the product of the number of docking sites, *N*, and of the release probability at each docking site, $P_i$ (see SV release normalized to *N*, *Figure 2C*). As our conclusions rested on the values of $P_i$, not $s_i$, we made a special effort to verify that our estimates of *N* did not include any systematic error.

One method to determine *N* is to perform a variance–mean analysis on $s_i$, plotted as a function of *i* in a given experimental condition (*Malagon et al., 2016*). In the case illustrated in *Figure 3A*, left (in the presence of 4-AP; same experiment as in *Figure 2A*), the fit indicated a value *N* = 3.8 ± 0.2, close to the whole number *N* = 4. We compared this result with that of an alternative method where the distribution of observed SV numbers at individual *i* values is fitted with a binomial distribution, leaving *N* as a free parameter (*Malagon et al., 2016*). When applying this method to the responses to the first two stimuli, the best fit was obtained with *N* = 4 (*Figure 3A*, center and right). Therefore, the two methods give the same estimate for *N* (*N* = 4 in the present case). Inverting the relation $s_i = NP_i$ with the value *N* = 4 gives $P_1$ = 0.86 and $P_2$ = 0.66 in this example.

Across experiments, *N* values returned by variance–mean curves did not change when adding 4-AP or after PTP induction (*Figure 3B*). Common *N* values across experimental conditions were consistent not only with variance–mean estimates, but also with binomial analysis of $s_1$, as illustrated in *Figure 3C* for an example experiment, and in *Figure 3D* for a group of 4 experiments with *N* = 4. Therefore, a single *N* value was adopted for each experiment. Because *N* is better constrained when *P* values are high, *N* was taken as the best whole value obtained from binomial analysis of $s_1$ and $s_2$ in 4-AP (see Methods). The invariable *N* values indicate that the addition of 4-AP and the PTP protocol do not increase the number of docking sites in the time frame of the experiments. *N* values determined with the binomial method in 4-AP varied among experiments from 2 to 6, with a mean value of 3.6 ± 0.3 (*n* = 14 experiments). In the subset of experiments illustrated in *Figure 3B*, the corresponding mean number was 4.2 ± 0.4 (range, 3–6), close to the mean value obtained by variance–mean analysis (4.3 ± 0.2, *Figure 3B* center; *n* = 8 experiments).

## In 4-AP and after PTP, the sum $P_1 + P_2$ is an estimate of maximal RRP size per docking site

*Figure 4* shows average values of $P_1$, $P_2$, and $P_1 + P_2$, across experimental conditions (in control, in 4-AP, and after PTP). $P_1$ significantly increased in 4-AP compared to control, and after PTP compared to the previous data in 4-AP (control: 0.39 ± 0.04; 4-AP: 0.69 ± 0.03, paired *t*-test p$_{pt}$ < 0.01 compared to control; PTP: 0.88 ± 0.04, p$_{pt}$ < 0.01 compared to 4-AP; *n* = 14 for control and 4-AP, *n* = 8 for comparison after and before PTP protocol; p$_{pt}$ calculated for paired results; *Figure 4*, left). $P_2$ significantly

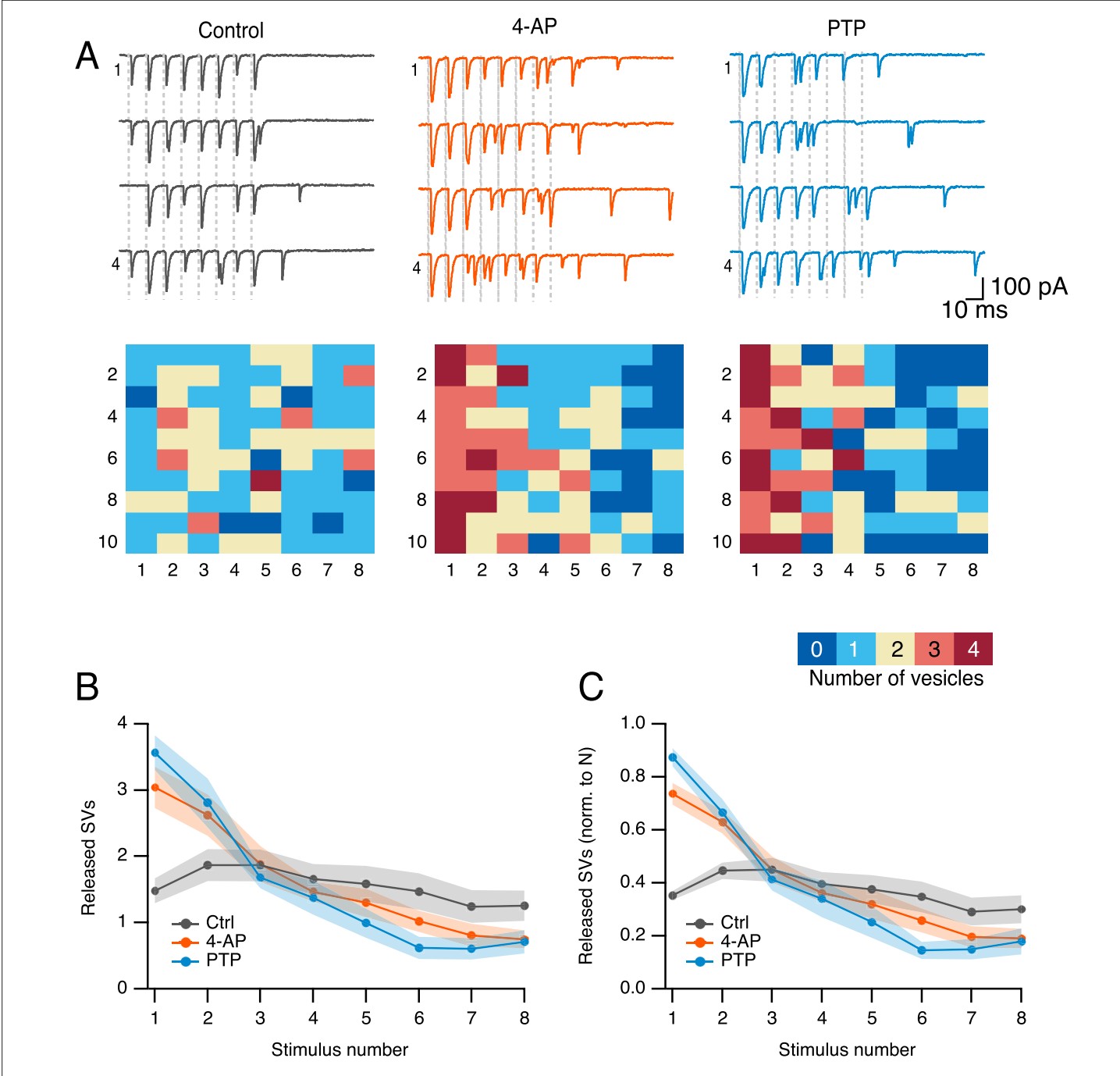

**Figure 2.** Simple synapse responses to 8-AP trains under conditions maximizing release probability and docking site occupancy. (**A**) Upper panels: Exemplar traces from a simple synapse recording performed in 3 mM external calcium concentration, before (left) or after (center and right) application of the potassium channel blocker 4-amidopyridine (4-AP). Each trace shows a response to an 8-AP stimulation train at 100 Hz (stimulation times indicated by vertical dotted lines). In 4-AP, a post-tetanic potentiation (PTP) protocol was applied to further enhance readily releasable pool (RRP) size (right). Lower panels: Tables showing in color code the number of synaptic vesicle (SV) released per stimulus, as a function of stimulus number (columns) and train number (rows; traces for rows 1–4 shown in upper panels). (**B, C**) Group results showing released SV numbers ($m$ ± standard error of the mean SEM) as a function of stimulus number, in control (3 mM external calcium, no further addition or manipulation, black), after addition of 4-AP (red), and when combining 4-AP with PTP induction (blue). SV numbers are given per synapse in (**B**). In (**C**), they are given per docking site, after normalization in each experiment with respect to docking site number. Number of experiments: $n = 8$.

The online version of this article includes the following source data for figure 2:

**Source data 1.** Released synaptic vesicle (SV) counts for individual experiments in various experimental conditions.

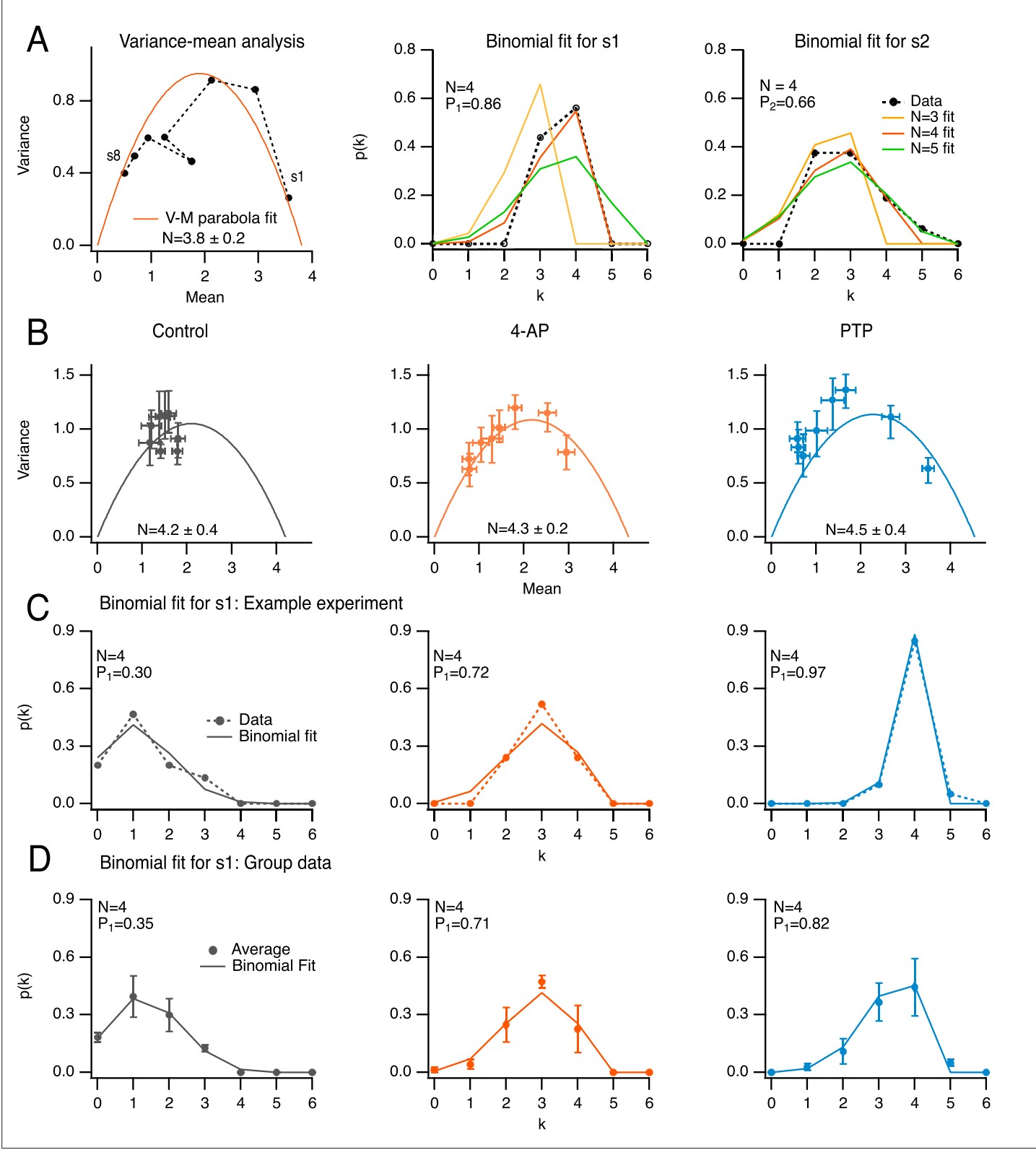

**Figure 3.** Determination of docking site number using variance/mean analysis and binomial distribution analysis. (**A**) Comparison of the determination of *N*, the docking site number, either with variance–mean analysis of released synaptic vesicle (SV) numbers for stimuli 1–8 (left; black closed circles: data; solid red trace: fit with parabola), or by fitting distributions of released SV numbers in response to the first (center) and second (right) stimulus (black closed circles: data; solid traces: fits with binomial models assuming various *N* values). Same experiment as in **Figure 2A**, center, in the presence

*Figure 3 continued on next page*

*Figure 3 continued*

of 4-amidopyridine (4-AP). The estimated *N* value with the variance–mean method was 3.8 ± 0.2 (left), and 4 with the binomial analysis (center and right, red curves). (**B**) Group analysis of eight experiments, normalized to four docking sites, showing variance–mean relation for stimuli 1–8 in control, in 4-AP and during post-tetanic potentiation (PTP; closed circles and error bars: *m* ± standard error of the mean [SEM]; solid curves: fit of means with parabola). (**C**) Exemplar experiment where *N* was determined as 4, showing a close similarity between distributions of released SV numbers in response to first AP in the three conditions (closed circles with dotted curves) and binomial fits assuming *N* = 4 (solid curves). (**D**) Group analysis of four experiments as in C (closed circles and error bars: *m* ± standard error of the mean [SEM]; solid curves: fit of means with binomial model assuming *N* = 4).

The online version of this article includes the following figure supplement(s) for figure 3:

**Figure supplement 1.** Lack of correlation between $P_1$ and *N*.

increased in 4-AP compared to control, and it remained stable after PTP (control: 0.46 ± 0.04; 4-AP: 0.64 ± 0.03, $p_{pt}$ < 0.01 compared to control; PTP: 0.67 ± 0.07, $p_{pt}$ > 0.05 compared to before protocol; *Figure 4*, middle).

As $p_r$ and the docking site occupancy before stimulus *i* ($\delta_i$) are probabilities, they are smaller than 1. Therefore, the relation $P_i = \delta_i p_r$ implies that both $\delta_i$ and $p_r$ are larger than $P_i$. Since $P_1$ is 0.69 in 4-AP, and 0.88 in PTP, both $\delta 1$ (noted as $\delta$) and $p_r$ are ≥0.69 in 4-AP, and ≥0.88 in PTP. These results indicate that we did reach the conditions of elevated release probability and docking site occupancy that are required for a meaningful evaluation of RRP size.

During PTP, virtually all SVs initially bound to the proximal docking site are released by the first AP, making replenishment of this site necessary for release in response to the second AP. $P_2$ then reflects the number of SVs that have moved from the distal (replacement) site to the proximal (docking) site during the time interval separating the first and the second AP (10 ms). In the RS/DS model, this number is determined by *r*, the probability of transition from replacement site to docking site. As this transition is known to occur within a few milliseconds after a strong calcium entry (*Miki et al., 2018*; *Miki et al., 2016*), *r* is likely elevated (possibly close to 1) during the 10-ms interval separating the first two APs. Apart from the *r* transition, obtaining a release event in response to the second AP also depends on the initial occupancy of the replacement site, $\rho$, as well on $p_r$. Altogether, in this condition, $P_2$ can be approximated by the product of $\rho$, *r*, and $p_r$. As the experimental value of $P_2$ in PTP is 0.67, by the same reasoning as above, it follows that both $\rho$ and *r* are ≥0.67.

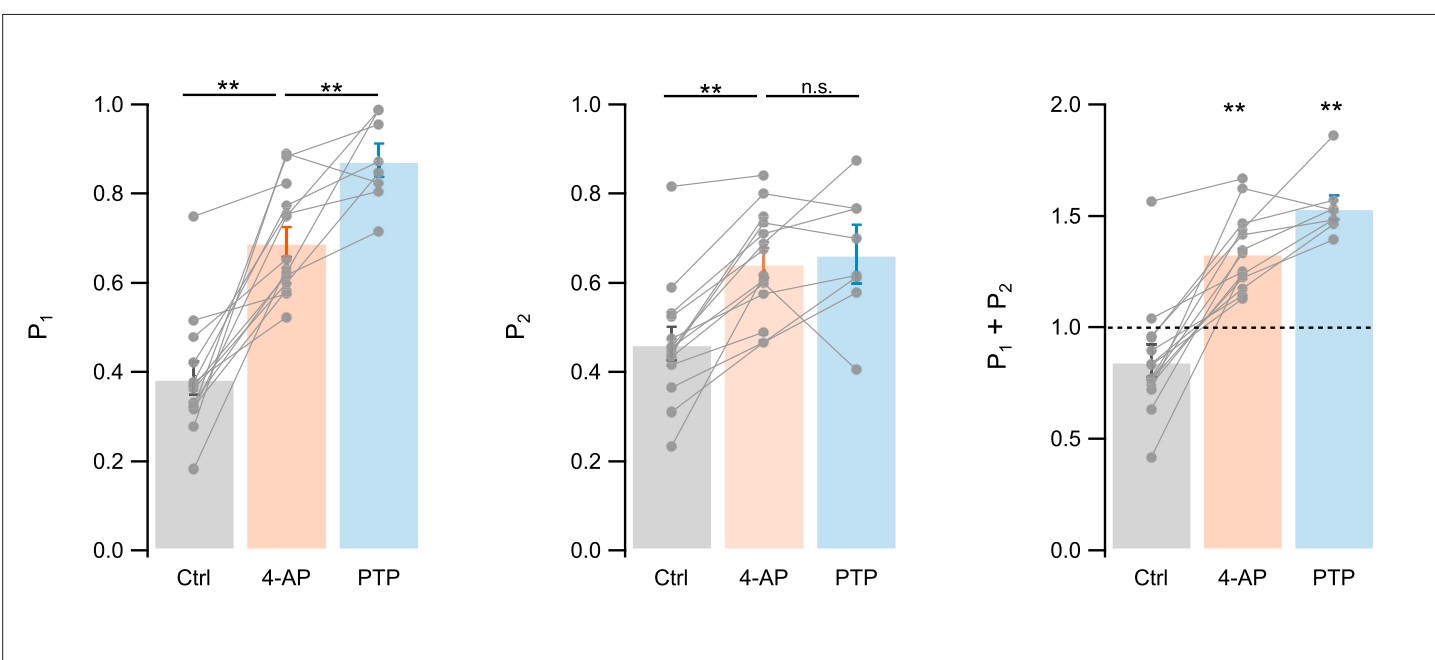

**Figure 4.** Summary results for $P_1$, $P_2$, and $P_1 + P_2$. Summary results for $P_1$ (left), $P_2$ (center), and $P_1 + P_2$ (right) in control, in 4-amidopyridine (4-AP), and during post-tetanic potentiation (PTP). Gray closed circles and linking lines indicate individual experiments; bars and associated margins indicate *m* ± standard error of the mean (SEM). *n* = 14 experiments for control and 4-AP, and *n* = 8 experiments for PTP. ** labels indicate significant differences between conditions (left and center, paired *t*-tests), and significant differences from 1 (right), with p < 0.01.

The above argument assumes that SV replenishment through the combined $s$ and $r$ pathway (SVs upstream of the RRP) does not significantly contribute to $P_2$. Consistent with this assumption, the steady-state response at the end of a train is low, indicating that the probability of entry into the replacement site, $s$, is low. Furthermore, the recruitment of upstream SVs requires an empty replacement site and will therefore be delayed until the replacement site is free. Because $\rho$ is close to 1 initially, the combined $s$ and $r$ pathway is unlikely to contribute significantly to $P_2$ (see numerical assessment of the potential error in Methods).

Altogether, our results show that in 4-AP and after PTP, the sum $P_1 + P_2$ can serve as an estimate of the maximal RRP size per docking site.

## The RRP can contain more than 1 SV per docking site

As shown in the right plot of *Figure 4*, the sum $P_1 + P_2$ was significantly larger than 1 both in 4-AP (1.34 ± 0.05; $p_t < 0.01$; $n = 14$) and during PTP (1.54 ± 0.05; $p_t < 0.01$; $n = 8$). This indicates that the initial RRP size was larger than 1 SV per docking site in both of these experimental conditions.

## Under high release probability conditions, synaptic depression, and asynchronous release are both increased as predicted by the RS/DS model

When examining the time course of release in our control conditions (3 mM $Ca_0$), we found that peak release rates decreased as a function of stimulus number at the end of AP trains, reflecting synaptic depression (*Figure 5A*). The extent of this depression grew markedly in the presence of 4-AP (second row) compared to control (first row). Depression became even more prominent after PTP (third row). In several experiments, after inducing PTP in the presence of 4-AP, we changed the frequency of stimulation to 200 Hz ($n = 5$). Under these conditions, synaptic depression was further enhanced to such an extent that individual peak release responses could not be distinguished at the end of the train (fourth row).

Previous work suggests that while synchronous release (calculated over a period of 5 ms following each AP: dashed traces in *Figure 5B*) is largely due to the fusion of docked SVs upon AP arrival, asynchronous release (yellow traces in *Figure 5A, B*) primarily originates from SVs that did not belong to the RRP at the time of the AP (*Miki et al., 2018*; *Miki et al., 2016*; *Sakaba, 2006*; *Tran et al., 2022*). Such SVs can be released at a low rate within the following inter-AP interval after moving through distal and proximal docking sites, provided that these sites are empty. Therefore, the proportion of asynchronous release gives an indication on the degree of filling of the RRP. In control, only synchronous release was observed at the beginning of AP trains, but a significant share of asynchronous release was observed near the end of the train (top panel of *Figure 5B*). In 4-AP and during PTP, the proportion of asynchronous release was initially as low as in control, but this proportion increased more rapidly as a function of stimulus number (second and third panels of *Figure 5B*). Strikingly, when increasing the stimulation frequency to 200 Hz during PTP, most release events near the end of the trains were asynchronous, and synchronous release was not detectable for $i > 5$ (bottom right panel in *Figure 5B*). Differences in release time courses between experimental conditions can be appreciated by examining superimposed cumulative release curves (*Figure 5C*). Altogether, the evolution of the synchronous component of release during trains (*Figure 5A*) and that of the proportion of synchronous vs. asynchronous release (*Figure 5B*) both indicate that RRP depletion during trains increases markedly from one condition to the next and becomes close to complete for 200 Hz stimulation during PTP.

According to the scheme of *Figure 1B*, in the extreme case where the RRP is full and $p_r = r = 1$, the RRP should be equally distributed between $s_1$ and $s_2$, with values close to 1 per docking site in both cases, while $s_i$ should be small for $i > 2$. The bottom right panel in *Figure 5B* depicts the evolution of synchronous release in the 200 Hz condition (recorded under 4-AP and after PTP induction). Here, $s_2$ is only slightly smaller than $s_1$, and $s_i$ falls off abruptly as a function of $i$ for $i > 2$. This fall-off can be fitted by an exponential function with an asymptote at 0 ordinate value, and a time constant of 1.66 inter-AP intervals (corresponding to 8.3 ms). These data indicate that synaptic depression is weak for $i = 2$, but very rapid and complete thereafter, thus approaching the predictions of the scheme of *Figure 1B*. To quantify the extent of synaptic depression in the various experimental conditions, we calculated the amount of synchronous release per AP at the end of the trains from the slopes of

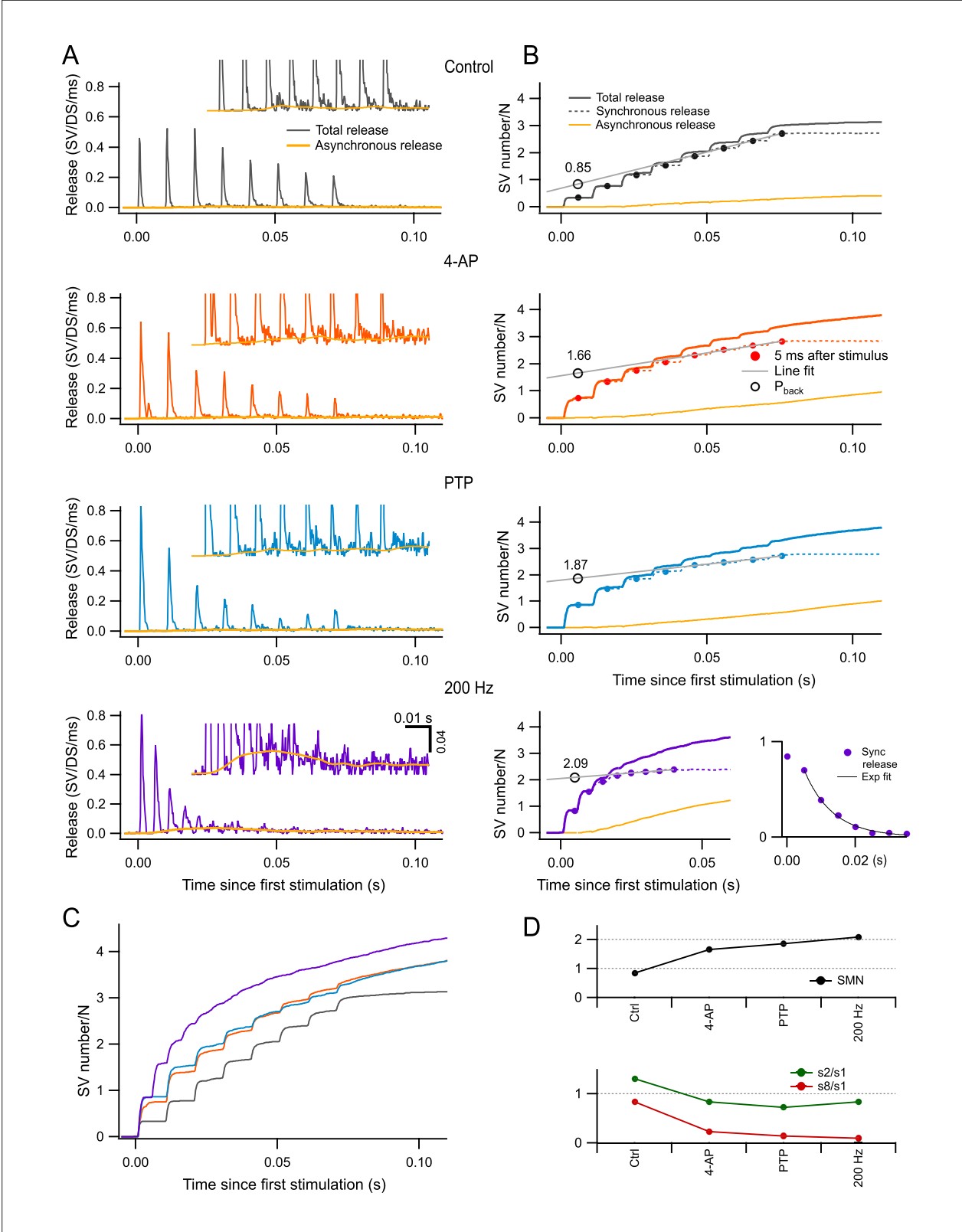

**Figure 5.** Estimation of readily releasable pool (RRP) size per docking site by back-extrapolation of cumulative release numbers. (**A**) Release rate per docking site (from top to bottom: black, control; red, 4-amidopyridine [4-AP]; blue, during post-tetanic potentiation [PTP]; purple, during PTP when accelerating the stimulation rate to 200 Hz) averaged from *n* = 5–7 experiments with 8-AP trains. Yellow curves close to the baseline indicate estimated asynchronous release in each plot. Insets show expansions of the plots in the low-amplitude range. (**B**) Cumulative numbers of released synaptic vesicles

*Figure 5 continued*

(SVs) obtained by integration of the plots in (**A**) (continuous black, red, blue, and purple curves: total release; dotted curves: synchronous release; continuous yellow curves: asynchronous release). Closed symbols indicate estimates of cumulative synchronous release per docking site as a function of stimulus number (*i*); they are placed 5 ms after each stimulus, when measurements were complete. A regression line through the dots obtained for *i* = 6–8 (black plot) or for *i* = 5–8 (red, blue, and purple plots) was back-extrapolated to the time of the first measurement, giving estimates of the RRP per docking site (open symbols: $P_{back}$). Bottom right panel: Plot of $s_i$ (synchronous release only) in conditions of 4-AP and PTP, at a stimulation frequency of 200 Hz. An exponential fit for *i* > 2 displays a time constant of 8.3 ms and a 0-ordinate asymptote. (**C**) Superimposition of the cumulative total release rates shown in (**B**), illustrating the gradual shift from synchronous to asynchronous release as a function of stimulus number in the various conditions of panels (**A, B**). (**D**) Plots of RRP size per docking site (upper panel), of the paired-pulse ratio ($s_2/s_1$, green points in lower panel), and of synaptic depression ($s_8/s_1$, red points in lower panel) calculated for synchronous release in the different conditions illustrated in (**A, B**). *n* = 7 independent experiments for 100 Hz stimulations (control, 4-AP, and PTP); five of these experiments included also data at 200 Hz after PTP.

regression lines fitted to the cumulative synchronous release during the steady state of each condition (gray lines in *Figure 5B*). We then divided this number by the number of SVs released by the first AP. The resulting curve steeply decreased from 0.83 in control conditions to 0.23, 0.14, and 0.10 in 4-AP, PTP, and 200 Hz, respectively (red curve in *Figure 5D*, bottom). This confirms the increasing effectiveness of the various conditions chosen in the present study to enhance synaptic depression. In contrast, the paired-pulse ratio, calculated as $s_2/s_1$, changed much less steeply across experimental conditions, and it remained close to 1 even when $s_8/s_1$ had reached very low values (green curve in *Figure 5D*, bottom). This indicates that the second response in a train is largely immune to synaptic depression, as predicted by the scheme in *Figure 1B*.

## RRP estimate obtained from summed released SV numbers

To obtain a more accurate estimate of the RRP size, we adapted the classical Schneggenburger–Meyer–Neher (SMN) method of back-extrapolation of cumulative excitatory postsynaptic potential (EPSC) amplitudes (*Schneggenburger et al., 1999*), using SV numbers in place of EPSC amplitudes. The SMN method is accurate only under conditions where the synapse is strongly depressing (*Neher, 2015*). As described above, synaptic depression and RRP depletion were marked under our experimental conditions. Therefore, we applied the SMN method to the synchronous release points obtained 5 ms after each AP (*Neher, 2015*; *Schneggenburger et al., 1999*; *Thanawala and Regehr, 2013*). Summed numbers of synchronously released SVs for stimuli 6–8 (in control conditions) or 5–8 (other conditions) were fitted to straight lines that were back-extrapolated to the time of stimulus number 1, giving a value $P_{back}$ (*Figure 5B*, gray lines and open circles). The estimated RRP was 0.85 SV per docking site in control, 1.66 SVs per docking site in 4-AP, and 1.87 SVs per docking site in PTP. Thus, both in 4-AP and during PTP, the SMN method indicates an RRP size that exceeds 1 SV per docking site. This confirms the earlier findings on the sum of $P_1 + P_2$ in the same conditions (*Figure 4*). Finally, when using 200 Hz stimulation, the SMN method gave an RRP estimate of 2.09 SVs per docking site (*Figure 5B*, bottom). The higher extrapolated value obtained for 200 Hz stimulation compared to 100 Hz was attributable to a more profound extent of synaptic depression, as discussed earlier. Notably, the values after PTP, both at 100 Hz and at 200 Hz, are close to the maximum value of 2 predicted by the RS/DS model (*Figure 5D*, top).

## Modeling SV release

To test the ability of the RS/DS model to account for the effects of 4-AP and PTP, we next performed Monte Carlo simulations of synaptic output during trains following established procedures (*Miki et al., 2016*; *Figure 6*). With each AP, the replacement site/docking site assembly, as shown in *Figure 6A*, evolved with transition probabilities of *s* (filling of empty replacement site), *r* (filling of empty docking site), and $p_r$ (release probability of an occupied docking site). The values of *s*, *r*, and $p_r$ were assumed constant throughout the AP train. The initial occupancy of the docking site, $\delta$, was treated as a free parameter and the initial occupancy of the replacement site, $\rho$, was set to 1. The parameters *s*, *r*, $p_r$, and $\delta$ were determined in each experimental condition (control, 4-AP, and PTP) by fitting to the mean number of released SVs after the first five APs. Near the end of the train, additional phenomena introduced significant deviations from the model, as discussed later.

In *Figure 6*, simulation and experimental results are compared in two different plots: the $s_i$ plot, showing the mean number of released SVs during a train (top row), and the cov($S_i$, $s_{i+1}$) plot, showing

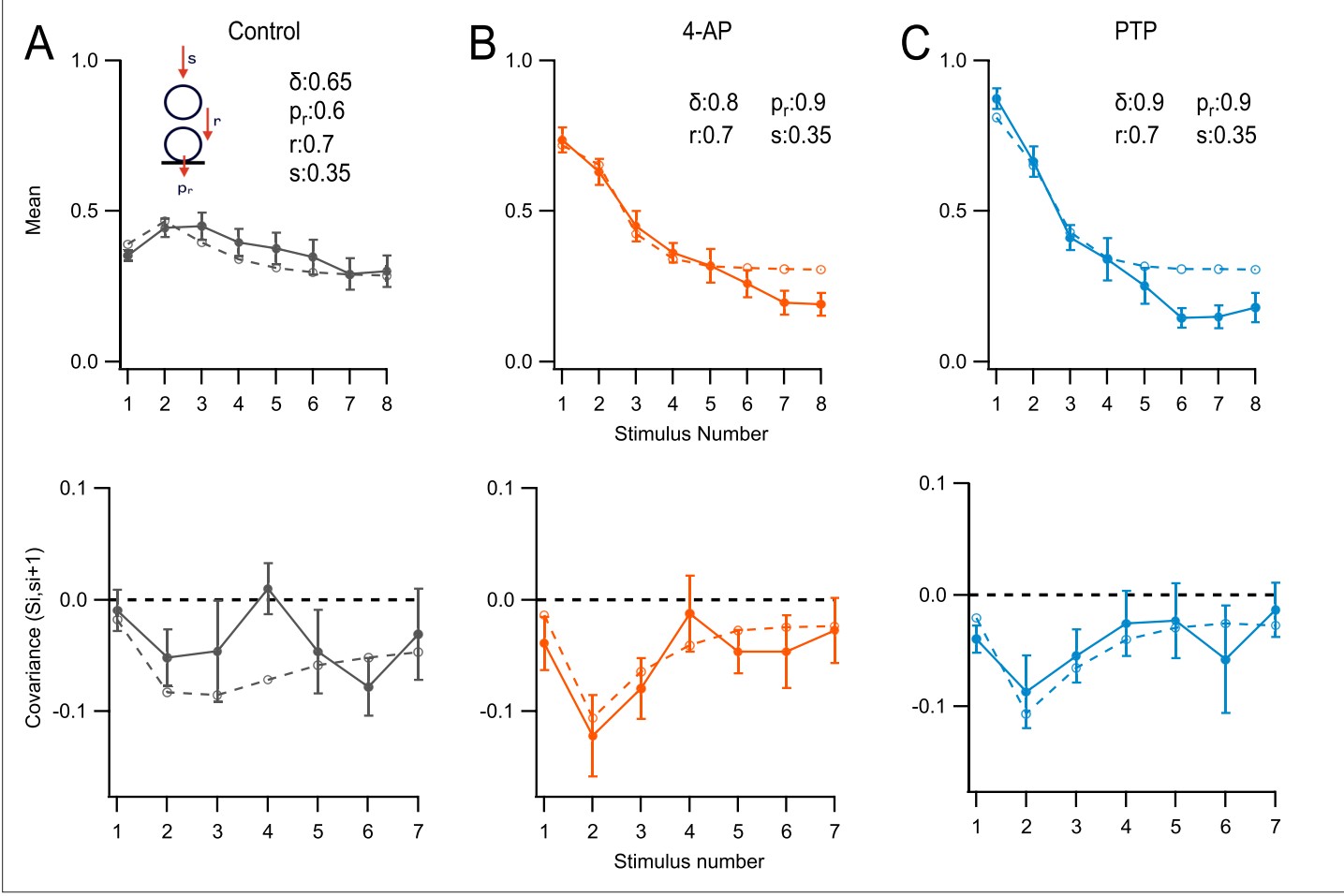

**Figure 6.** Modeling. Monte Carlo simulations of the responses to an 8-AP train at 100 Hz for a simple synapse obeying the replacement site/docking site (RS/DS) two-step model. Here, $s_i$ values represent numbers of released synaptic vesicles (SVs) per docking site after stimulus number $i$, uncorrected for asynchronous release. $S_i$ values represent the sum of $s_i$ up to stimulus number $i$. Experimental data ($m \pm$ standard error of the mean [SEM]) are displayed as closed symbols, and model results as open symbols. (**A**) Mean SV release number ($<s_i>$ vs. $i$: upper panel), and covariance of $S_i$ and $s_{i+1}$ lower panel: covar($S_i$, $s_{i+1}$) vs. ($i$) in control conditions. (**B**) As in (**A**), in 4-amidopyridine (4-AP). (**C**) As in (**B**), during post-tetanic potentiation (PTP).

The online version of this article includes the following figure supplement(s) for figure 6:

**Figure supplement 1.** Replacement site/docking site (RS/DS) model with finite intermediate pool (IP).

**Figure supplement 2.** Replacement site/docking site (RS/DS) model vs. loose state/tight state (LS/TS) model.

the covariance between the total number of released SVs up to stimulus $i$ and the number of released SVs after the following stimulus (bottom row; *Miki et al., 2016*).

In control conditions, the model returned an initial docking site occupancy $\delta = 0.65$. The release probability of docked SVs, $p_r$, was 0.6. The probability to fill an empty replacement site during one inter-stimulus interval, $s$, was 0.35. Finally, the transition probability from replacement site to docking site during one inter-stimulus interval, $r$, was 0.7. With these control values, the model was able to capture many features of the evolution of mean numbers and covariance as a function of stimulus number (*Figure 6A*).

As expected, the model indicated a markedly increased probability of release of a docked SV in the presence of 4-AP, from 0.6 to 0.9. $\delta$ increased moderately (from 0.65 to 0.8), possibly because the Ca$^{2+}$ concentration increased in the presynaptic terminals (*Malagon et al., 2020*). The other two kinetic parameters $r$ and $s$ retained their control values (*Figure 6B*). During PTP, model parameters were kept as before, except that the docking site occupancy increased from 0.8 to 0.9 (*Figure 6C*).

## The RS/DS model reproduces well the covariance of SV release counts

The model derived from the fit of $s_i$ curves was used without further parameter adjustment to fit covariance curves. As may be seen in the lower panels of **Figure 6**, the model correctly predicted the general amplitude and the shape of the $cov(S_i, s_{i+1})$ plot as a function of $i$ in the three experimental conditions of **Figure 3**. Covariance results and simulations were rather shapeless as a function of $i$ in control, but they displayed a marked peak for $i = 2$ in 4-AP and in PTP.

The initial low covariance value in 4-AP and PTP ($cov(S_1, s_2) = cov(s_1, s_2)$) can be explained as follows. As $p_r$ is very high ($p_r = 0.9$ according to the simulation shown in **Figure 6**, upper panels), successful release in response to the first AP mainly depends on the docking site being occupied prior to that AP. Following release elicited by this AP, almost all docking sites are empty. They refill during the first inter-AP interval with a probability $r$, independently of the initial state of occupancy prior to the first AP. Therefore, the variations in $s_2$ primarily originate in variations in the docking site occupancy prior to the second AP, and they are almost entirely independent of the variations in $s_1$. This results in a low-amplitude negative value for the first entry in the covariance plot.

Let us now consider the second point of the covariance plot, $cov(S_2, s_3) = cov((s_1 + s_2), s_3)$. Following the second AP, some sites have released their entire RRP during the first two APs, following a successful $r$ transition during the first inter-AP interval. These sites tend to have a high $(s_1 + s_2)$ value and a low $s_3$ value. Other sites have retained their original SV on the replacement site up to the second AP, as the $r$ transition has failed to occur during the first inter-AP interval. These other sites tend to have a low $(s_1 + s_2)$ value and a high $s_3$ value. This compensatory effect between $(s_1 + s_2)$ and $s_3$ results in a high-amplitude negative value for the second entry in the covariance plot.

## Correcting for the depletion of an upstream intermediate pool

We have found recently that replacement SVs are fed by an upstream vesicular pool of small size, the intermediate pool (IP: **Tran et al., 2022**). Near the end of a train, as the IP is depleted, RRP replenishment is reduced, so that the synaptic output is also reduced. Accordingly, the deficit of synaptic output observed for $i = 6–8$ in **Figure 6** is likely explained by a depletion of the IP. To explore this possibility further, we performed additional simulations assuming an IP with an initial pool size of 10–15 SVs for an AZ of four docking sites, corresponding to 2.50–3.25 SVs per docking site. The IP was replenished by an infinite recycling pool with a transition probability of $r_{IP}$. Coupling between IP size and RRP was introduced by making the transition probability $s$ proportional to the filling state of the IP. This resulted in an improvement of the fit near the end of the train (**Figure 6—figure supplement 1**).

After performing the IP correction, a residual discrepancy remained between simulated and experimental data near the end of AP trains. To explain this discrepancy, we note that our model does not include a provision accounting for asynchronous release. At the end of AP trains, asynchronous release reflects the movement of replenishment SVs across the RRP without stopping at RS or DS sites, due to persistent elevation of the presynaptic calcium concentration (**Miki et al., 2018**). Our model artificially forces these SVs to stop at RS and/or DS sites and to be counted as synchronous release. This causes synchronous release to appear larger in the model than in experimental data near the end of trains, even after IP correction.

## Comparison between RS/DS and LS/TS models

We next compared the predictions between the RS/DS model (without consideration of the IP) and the LS/TS model (**Figure 6—figure supplement 2**). In the LS/TS model, as in the RS/DS model, we optimized parameters to fit the first five $s_i$ values. In control conditions, both RS/DS and LS/TS models gave reasonably close representations of the data (**Figure 6—figure supplement 2B**). Note that the LS/TS model used very high values of $r$ (0.95) and $s$ (0.8).

In 4-AP, the LS/TS model could account for $P_1$ by increasing $\delta$ and $p_r$, but it displayed a value of $P_2$ below experimental observations (**Figure 6—figure supplement 2C**, upper panel). A variation on the LS/TS model adds a transitory state, the tight labile state, which is present only at high frequencies (**Lin et al., 2022**). It consists of LS vesicles that tightly dock for a short time after an AP. This can be understood as a short-lived increase in $r$. In our model such a modification cannot increase synaptic output significantly, as $r = 0.95$. Introducing an IP with a size of 10 improved the fitting only marginally (upper panel in **Figure 6—figure supplement 2C** ).

The LS/TS model could account for cov($S_i$, $s_{i+1}$) data in control conditions (*Figure 6—figure supplement 2B*, lower panel), but it failed to do so in 4-AP (*Figure 6—figure supplement 2C*, lower panel). In 4-AP, whereas data and RS/DS simulations display a peak amplitude for $i = 2$, the LS/TS simulation displays a peak amplitude for $i = 1$. In the LS/TS model, release events following the first AP come from sites initially in the TS. Such release events are unlikely to be followed by an immediate TS refilling from the LS, since LS occupancy is incompatible with TS occupancy. Conversely, most release events following the second AP are due to sites that were in the LS state before the first AP, and were therefore unable to release in response to the first AP. This results in a large amplitude negative covariance point for $i = 1$. The covariance then decreases for higher $i$ values as the RRP is depleted and release increasingly relies on replenishment through the s pathway.

These simulations suggest that the LS/TS model, which is simpler than the RS/DS model and contains less free parameters, can be considered as a good approximation for synapses where release probability and/or docking site occupancy are moderate.

## Discussion

Our estimate of RRP size per docking site is 1.66 SVs in 4-AP and 1.87–2.09 SVs in 4-AP after PTP (at 100 and 200 Hz stimulation rates, respectively; *Figure 5D*). These results suggest a maximum RRP size of two SVs per docking site at PF–MLI synapses, in agreement with the predictions of the RS/DS model (*Figure 1A*).

### Functional evidence in favor of the RS/DS model at PF–MLI synapses, and remaining uncertainties on the molecular composition and morphological arrangement of docking sites

Previous studies have led to the proposal of the RS/DS model at PF–MLI synapses based on (1) variance–mean analysis of SV counts in response to AP trains; (2) variance–mean analysis of cumulative SV counts; (3) covariance analysis of SV counts for successive AP stimulations in a train; and (4) two-step release during trains (*Miki et al., 2016*; *Miki et al., 2018*). The present study adds to this evidence by extending covariance data to conditions of very high docking site occupancy and release probability, as discussed above, and by showing that the ratio of RRP size to number of docking site reaches a maximum of 2 under the same conditions. Taken together these studies build a strong case in favor of the RS/DS model in the present preparation from a functional point of view.

However, uncertainties remain concerning the molecular composition and the morphological arrangement of docking sites. This is particularly true for PF–MLI synapses. Disrupting actin filaments by latrunculin B or inhibiting myosin II by blebbistatin results in an uncoupling between RS and DS (*Miki et al., 2016*). Activating synapsins following PKA activation increases the RRP size without changing $p_r$ (*Vaden et al., 2019*). Apart from these findings, the effects of specific molecular modifications on the RS/DS system of PF–MLI synapses remain to be explored. Likewise, morphological data indicating SV movement between RS and DS are lacking in this preparation, although such data have been found in other preparations (*Chang et al., 2018*; *Kusick et al., 2020*; *Miki et al., 2020*; *Martín et al., 2020*; *Vandael et al., 2020*; *Fukaya et al., 2023*). Further molecular and morphological studies are needed to ascertain the applicability of the RS/DS model to the PF–MLI synapse.

### Assumptions behind our analysis, and possible limitations of our conclusions

Our analysis and conclusions rest on several critical assumptions that are examined in turn below.

A first assumption of our RS/DS model is that within an AZ, all RS/DS assemblies operate in a similar manner. Alternative models would hold that release sites are not equivalent, and that some sites release SVs with much higher $p_r$ values than others (parallel models: *Eshra et al., 2021*; *Hallermann et al., 2010*; *Ritzau-Jost et al., 2014*; *Ritzau-Jost et al., 2018*). In general, distinguishing between parallel and sequential models is a difficult task that cannot be solved with absolute certainty. A study of mean and variance results at the synapse between PFs and Purkinje cells (PF–PC synapses) suggested at first a parallel model for this synapse (*Valera et al., 2012*), but additional results from the same group examining the behavior of PF–PC synapses in response to long trains at various stimulation frequencies favored a sequential model instead (*Doussau et al., 2017*). At PF–MLI synapses,

our previous study of variance–mean data indicated a sequential model (*Miki et al., 2016*). Based on this previous work, we framed the interpretation of the present results on sequential models (RS/DS or LS/TS) with homogeneous docking/release properties; but alternative parallel models cannot be ruled out entirely.

A second assumption is that $p_r$ does not change during a train, that is, $p_r$ is independent of $i$. In other synapses, a moderate increase of calcium entry occurs as a function of $i$, accounting for part of facilitation (e.g., in the calyx of Held: *Borst and Sakmann, 1998*; *Lin et al., 2022*). In contrast, at synaptic boutons of cerebellar GCs, AP-driven calcium entry is independent of $i$ (*Miki et al., 2016*; *Kawaguchi and Sakaba, 2017*). In addition, since our results and analysis indicate that $p_r$ is high in the present work (with an estimated value of 0.9 in 4-AP), any potential further increase of $p_r$ during trains is bound to be small.

A third assumption is that RRP release is kinetically separate from RRP replenishment from upstream pools (recycling pool and IP). In the SMN analysis, RRP replenishment is assumed to be linear as a function of time, and to correspond to the late linear part of the plot of cumulative release as a function of time. The initial rate of SV release calculated for the first two stimuli in the cumulative plot of *Figure 5C* in 200 Hz conditions is markedly higher than that calculated for the corresponding late period asymptotic line (with ratios of 2.6 in 4-AP, 3.3 in PTP, and 4.3 in 200 Hz), suggesting a good separation between release and replenishment. The actual ratio is likely larger than suggested by these numbers, since the rate of RRP replenishment is slower at the onset of the stimulation train (due to the cumulative calcium rise driving replenishment yet to reach its peak and the initial high filling state of the RRP inhibiting replenishment: *Ruiz et al., 2011*; *Thanawala and Regehr, 2013*; *Neher, 2015*). We calculated the contribution of replenishment to $s_1$ and $s_2$ release and found that it is negligible even under high $p_r$ conditions (see Methods). The large extent of synaptic depression of synchronous release obtained in 4-AP, after PTP and when using 200 Hz stimulation (*Figure 5D*), is consistent with a good separation between RRP release and replenishment.

Altogether, departures from the above assumptions (homogeneous release site properties inside an AZ; constant $p_r$ during a train; kinetic separation between RRP release and replenishment) cannot be ruled out, but such departures, if they occurred, are unlikely to have affected the main conclusions of our study.

## Our approach to measure the maximal RRP size: maximizing $\rho$, $\delta$, and $p_r$

While the RS/DS model had been previously proposed in this preparation (*Miki et al., 2016*; *Miki et al., 2018*), a key model prediction that the RRP size per docking site could reach a value close to 2 SVs had not been tested until the present work. An early estimate based on the SMN method gave a value close to 1 SV per docking site (*Miki et al., 2016*); this value was obtained in rats in a 3-mM external calcium concentration without K$^+$ channel blocker, leaving open the possibility for alternative models such as the LS/TS model. Likewise, in the present study the SMN method gives an estimate of 0.85 SV per docking site for mice in the same solution (*Figure 5B*, top). The question remained open as to whether larger values could be obtained under certain experimental conditions.

Importantly, the SMN method actually measures the difference between the original size of the RRP and the size reached at steady state (*Neher, 2015*). Because RRP depletion is incomplete during sustained AP stimulation, the extrapolation measurement yields an underestimate of the RRP size (*Neher, 2015*). In addition, the original RRP size does not necessarily correspond to the number of available SV-binding sites because at rest, a fraction of these sites is not occupied (*Miki et al., 2016*; *Neher and Brose, 2018*; *Trigo et al., 2012*). Altogether, because of incomplete RRP depletion and because of incomplete site filling, traditional RRP measurements return values that can be markedly smaller than the number of available binding sites in the RRP. In order to perform a meaningful RRP size measurement for the purpose of the present work, it appeared important to maximize release probability (and consequently, synaptic depression) and to maximize site occupancy.

No special step was taken to increase $\rho$ since earlier studies in rats suggested high $\rho$ values (0.9: *Miki et al., 2018*; *Miki et al., 2016*). Consistent with these previous studies, high $\rho$ values were needed in the present work to explain high $P_2$ values under conditions of very high release probability (in 4-AP after PTP). Our simulations indicate $\rho$ values indistinguishable from 1 in all experimental conditions (control, 4-AP, and PTP). In all three cases, $\rho$ estimates were larger than $\delta$ estimates. These results indicate that in basal conditions, the rate constant of the undocking transition (from docking

site to replacement site) is larger than that of the docking transition (from replacement site to docking site).

Previous work indicates that elevating the external calcium concentration enhances the RRP size (**Thanawala and Regehr, 2013**) and more specifically, increases $\delta$ (**Blanchard et al., 2020**; **Kobbersmed et al., 2020**; **Malagon et al., 2020**; **Tanaka et al., 2021**). Therefore, all experiments were carried out in 3 mM external calcium. Under these conditions, our $\delta$ estimate is 0.7 (**Figure 6**), higher than our previous estimates (0.45–0.47: **Malagon et al., 2020**; **Miki et al., 2016**). The difference is likely due to species difference (mice in the present work, vs. rats in previous studies), since the estimate by **Miki et al., 2016** rested on a similar approach to that of the present work. Thus, selecting mice rather than rats for the present experiments, together with using a high external calcium concentration, contributed to enhancing docking site occupancy. In addition, we were able to further increase $\delta$ (from 0.7 to 0.9 according to the analysis of **Figure 6**) by applying a PTP protocol. This result is consistent with previous work at the calyx of Held (**Lee et al., 2010**) and at hippocampal mossy fiber synapses (**Vandael et al., 2020**), as well as with our recent finding that at PF–MLI synapses, PTP increases SV docking of a specific subclass of SVs originating from a local, calcium-sensitive recycling pathway (**Tran et al., 2023**).

Finally, to maximize $p_r$, we combined the effects of the potassium channel blocker 4-AP with those of the high external calcium concentration. In agreement with earlier results in rats (**Malagon et al., 2020**), this combination allowed us to reach $p_r$ values close to 1 (estimated at 0.9 in 4-AP both before and after PTP induction: **Figure 6**). Combining large values of $\delta$ and $p_r$ in the present work led to a $P_1$ value close to 1 during PTP ($P_1 = 0.88$, compared to a maximum $P_1$ value of 0.51 in our previous work: **Malagon et al., 2020**). Obtaining a $p_r$ value close to 1 was important here as it ensured that initially occupied docking sites would be cleared after a single AP, thus freeing the way for the docking of SVs initially in the replacement site during the following inter-stimulus interval.

## Reliability of our *N* estimates

Our conclusions depend on the accuracy of our *N* estimates. *N* estimates in the present work are particularly reliable because (1) SV counting eliminates potential errors linked to receptor saturation or desensitization in classical EPSC studies and (2) high P conditions make N estimates via variance–mean analysis or binomial analysis accurate (**Malagon et al., 2020**).

Even though high P conditions result in accurate *N* determination, one possible pitfall could be that under these conditions, certain SV release events could be too close in time to be distinguished, leading to $s_i$ undercount. Undercounting should worsen as release probability or docking site number grow, and to be worst for $i = 1$. $s_i$ undercount could lead to left shifts of variance–mean and binomial probability plots and hence to *N* underevaluation. Against this possibility, our estimates of the numbers of overlapping events at short intervals indicate a small extent of undercounting even under conditions of high release probability (see Methods). Secondly, our *N* estimates obtained either with variance–mean analysis (**Figure 3B**) or with binomial analysis (**Figure 3C, D**) were independent of release probability. Finally, undercounting at high P values predicts that apparent values of $s_1$ should grow sublinearly as a function of the estimated N value, while apparent $P_1$ values should increase as a function of N. Contrary to this prediction, we found that $s_1$ values increased linearly as a function of N, and that our $P_1$ estimates were independent of N (**Figure 3—figure supplement 1**). Altogether, it appears that undercounting did not significantly affect our *N* estimates.

## RRP size in various experimental conditions

Our results indicate that a single docking site can be associated with up to 2 SVs at simple PF–MLI synapses, thus giving strong experimental support in favor of the RS/DS model. Nevertheless, it is important to stress that under physiological conditions, the actual RRP size is likely significantly less than 2 SVs per docking site. In our control conditions, with 3 mM Ca$_o$, we obtained an estimate of 0.85 SV per docking site with the SMN method. This value is an underestimate since the residual synaptic responses were significant near the end of the AP train for these data (**Figure 5A**, top), indicating incomplete RRP depletion (**Neher, 2015**). A more realistic estimate based on the sum of $\rho$ and $\delta$ values obtained in our simulations for this condition is 1.65 SVs per docking site (with estimated values of 1 and 0.65 for $\rho$ and $\delta$, respectively: **Figure 6**). After PTP induction, corresponding estimates are 2.09 SVs per docking site (SMN method: **Figure 5D**, top panel) and 1.9 SVs per docking site (sum of $\rho$

and $\delta$ in relevant simulations, see *Figure 6*). Earlier studies in the rat suggested a value of 0.9 for $\rho$, independently of Ca$_o$, and a value $\delta = 0.2$ in 1.5 Ca$_o$ and $\delta = 0.5$ in 3 Ca$_o$, suggesting RRP sizes of 1.1 SVs and 1.4 SVs per docking site, respectively (*Malagon et al., 2020*; *Miki et al., 2016*). These results suggest that the RRP size varies across species and across experimental conditions. They further indicate that synapses do not operate at full RRP capacity under ordinary physiological conditions. Finally, our results together with previous work suggest that synapses modulate their RRP size during certain forms of synaptic plasticity, as discussed above for the case of PTP.

## Maximum RRP size for other synaptic types

Central mammalian synapses are very diverse functionally, and this is reflected in differences in the structure and molecular composition of their AZs (*Sakamoto et al., 2022*). PF–MLI synapses, as well as other synapses of the cerebellar molecular layer involving PF signaling (including mossy fiber-granule cell synapses and PF–Purkinje cell synapses), are characterized by a high level of synaptic facilitation as well as by fast SV replenishment (*Atluri and Regehr, 1998*; *Bao et al., 2010*; *Hallermann et al., 2010*; *Ritzau-Jost et al., 2014*; *Saviane and Silver, 2006*). In contrast, other synapses such as the calyx of Held display less facilitation and slower SV replenishment (*Taschenberger et al., 2005*; *Taschenberger et al., 2016*; *Lin et al., 2022*). At synapses between hippocampal mossy fibers and CA3 interneurons, where synaptic depression is prominent, the RRP size per docking site was recently estimated with the SMN method at 1.6 SVs in 3 mM Ca$_o$ (*Tanaka et al., 2021*), compared to 0.85 SV in the present study. Based on these results and on the simulations in the present work, it can be speculated that synapses with high p values exhibiting synaptic depression may more readily reach conditions where the ratio of RRP size to $N$ is larger than 1, compared with synapses with low p values exhibiting synaptic facilitation. Future work will be needed to test this prediction.

## Relation with EM studies

The finding of 2 SVs per docking site implies the existence of two separate SV-binding sites able to accommodate two SVs simultaneously. In EM studies performed at amphibian (*Szule et al., 2012*) and mammalian (*Nagwaney et al., 2009*) neuromuscular junctions, as well as in hippocampal cultured neurons (*Cole et al., 2016*), two sets of SVs likely belonging to the RRP have been identified. These EM data are consistent with the suggestion that the RRP comprises two classes of SVs that bind at distinct distances from release sites, with distance differences on the order of one SV diameter. In such a situation an RRP size of 2 per release site is plausible. Other EM studies identified a separate set of SVs located at a distance of about 5–10 nm from the plasma membrane. For these SVs, docking appears to involve a short distance movement from their pre-docked resting state to a fully docked state where they are in direct contact with the plasma membrane (*Chang et al., 2018*). This second set of EM results is consistent with the suggestion of two mutually exclusive pre-docked and docked states (*Neher and Brose, 2018*). In such a situation each docking site can accommodate maximally one SV, such that the maximal RRP size is 1 per docking site (LS/TS model, *Neher and Brose, 2018*; *Figure 6—figure supplement 2*). Altogether, some EM results appear to favor the RS/DS model (RRP size up to 2 SVs per docking site) while other EM results appear to favor the RS/DS model (RRP size up to 1 SV per docking site). It is possible that these differences reflect differences in synapse structure depending on synaptic class.

## Potential functional advantages of having up to 2 bound SVs per docking site

Various potential functional advantages can be envisaged for having up to 2 bound SVs per docking site.

Firstly, the RRP capacity is increased, since the RRP size can reach up to 2 SVs per docking site, compared to up to 1 SV per docking site in the LS/TS model. The increased RRP capacity translates into a larger responsiveness of the synapse at the onset of AP train stimulation. This is potentially important for simple synapses, having but a few docking sites, as a larger total number of released SVs improves the reliability of synaptic output during the first APs in a train.

Secondly, a mechanism of facilitation based on a calcium-dependent increase in docking occupancy (*Miki et al., 2016*; *Neher and Brose, 2018*; *Silva et al., 2021*) is more feasible if each docking

site can accommodate up to 2 SVs compared to 1 SV. This is because the complement of SVs available to refill the docking sites is larger in the former case than in the latter (*Miki et al., 2016*).

Third, doubly occupied docking sites contribute to spreading the release of the RRP over a wider range of AP numbers. For docking sites that are doubly occupied at train onset, the movement of SVs from replacement site to docking site must await clearing of the docking site by exocytosis, thus delaying the release of the SV initially occupying the replacement site to later AP numbers in the train. This effect, together with facilitation, counterbalances the synaptic depression following release of docked SVs, and helps retaining a relatively stable synaptic output as a function of AP number during trains.

A fourth potential functional advantage of doubly occupied docking sites is that the replacement site can act as an entry gate into the RRP. A high occupancy of the replacement site may result in a reduction of the flow of SVs into the RRP, independently of the state of occupancy of the docking site. By controlling the filling state of a separate replacement site, in ways that remain to be investigated, the synapse may be able to control RRP replenishment rate.

## Materials and methods
### Experimental model
C57BL/6 mice (10–16 days old; either sex) were used for preparation of acute brain slices. Animals were purchased from Janvier Laboratories (RRID:MGI:2670020). They were housed and cared for in accordance with guidelines of Université Paris Cité (approval no. D 75-06-07; animal rearing service of the BioMedTech Facilities, INSERM US36, CNRS UMS2009).

### Slice preparation
200-µm thick sagittal slices were prepared from the cerebellar vermis as follows. Mice were decapitated and the cerebellar vermis was carefully removed. Dissections were performed in ice-cold solution containing the following (in mM): 130 NaCl, 2.5 KCl, 26 $NaHCO_3$, 1.3 $NaH_2PO_4$, 10 glucose, 1.5 $CaCl_2$, and 1 $MgCl_2$ (artificial cerebrospinal fluid or ACSF). After cutting with a vibratome (VT1200S; Leica), slices were incubated in ACSF at 34°C for at least 45 min before being used for experiments.

### Synaptic electrophysiology
Whole-cell patch-clamp recordings were obtained from MLIs (comprising both basket and stellate cells). The extracellular solution contained (in mM): 130 NaCl, 2.5 KCl, 26 $NaHCO_3$, 1.3 $NaH_2PO_4$, 10 glucose, 3 $CaCl_2$, and 1 $MgCl_2$ (pH adjusted to 7.4 with 95% $O_2$ and 5% $CO_2$). The internal recording solution contained (in mM): 144 K-gluconate, 6 KCl, 4.6 $MgCl_2$, 1 EGTA (Ethylene glycol-bis tetraacetic acid), 0.1 $CaCl_2$, 10 HEPES (4-(2-hydroxyethyl)-1-piperazineethanesulfonic acid), 4 ATP-Na, 0.4 GTP-Na (pH 7.3, 300 mosm/l). All recordings were done at 32–34°C, after adding gabazine (3 µM) to block $GABA_A$ receptors. To establish a single GC–MLI connection, puffs using a glass pipette filled with the high $K^+$ internal solution were applied to identify a potential presynaptic GC. The same pipette was then used for extracellular electrical stimulation of the GC, with minimal stimulation voltage (10–40 V; 150 µs duration). If the resultant EPSCs, particularly those that occurred after a short stimulation train, were homogeneous, then it was likely that only one presynaptic cell was stimulated.

Recordings were only accepted as a simple synapse recording after analysis if the following three criteria were satisfied (*Malagon et al., 2016*): (1) the EPSC amplitude of the second release event in a pair was smaller than that of the first, reflecting activation of a common set of receptors belonging to one postsynaptic density; (2) all the EPSC amplitudes follow a Gaussian distribution with a coefficient of variation <0.5; and (3) the number of release events during the baseline recording was stable.

A broad-spectrum blocker of voltage-dependent $K^+$ channels, 4-aminopyridine (4-AP), was used at a final concentration of 15 µM to enhance the release probability (see below). 4-AP was dissolved in water and stock aliquots were kept at −20°C.

## Maximizing RRP measurement at simple synapses with a combination of high external calcium concentration, potassium channel blocker, and PTP

It was recently shown that increasing the external calcium concentration leads both to an increase in the occupancy of docking sites at rest, $\delta$, and to an increase of the release probability of docked SVs, $p_r$ (*Blanchard et al., 2020*; *Eshra et al., 2021*; *Kobbersmed et al., 2020*; *Malagon et al., 2020*; *Tanaka et al., 2021*). Both effects should help minimizing the difference between the number of SVs released after two APs and the number of SV-binding sites. Therefore, in the present work, we used an elevated external calcium concentration (3 mM). We also used the potassium channel blocker 4-AP, which is very effective in increasing release probability at various synapses including PF–MLI synapses (*Heuser et al., 1979*; *Ishikawa et al., 2003*; *Malagon et al., 2020*; *Tan and Llano, 1999*). In addition, PTP has been reported to increase the proportion of SVs in the fast releasing pool at the calyx of Held (*Lee et al., 2010*). At hippocampal mossy fiber synapses, PTP involves increases in the RRP size and in the number of docked SVs (*Vandael et al., 2020*). We likewise found an increase in SV docking during PTP at PF–MLI synapses (*Tran et al., 2023*). Based on these results, we used in the present work a PTP protocol to increase docking site occupancy ($\delta$), starting in a solution containing 3 mM external calcium and 4-AP. PTP was performed by applying 60 consecutive trains of 8 APs at 100 Hz, with an inter-stimulus interval of 2 s (*Tran et al., 2023*).

Finally, we used high stimulation frequencies (100–200 Hz) to maximize synaptic depression and to separate as much as possible RRP consumption from RRP replenishment (rate $s$ in *Figure 1A*; *Schneggenburger et al., 1999*).

### Quantification and statistical analysis

Offline analysis of electrophysiological data was performed using Igor Pro (WaveMetrics). Statistical analysis was conducted using Igor Pro. Data are expressed as mean ± standard error of the mean (SEM). Error bars in all graphs indicate SEM. Details of statistical tests are described in the text or figure legends. Statistical significance was accepted when $p < 0.05$. The number of cells tested is indicated by a lowercase $n$, each derived from a different animal. Within each cell, data were averaged in each experimental condition across a number of trials that was 20 or larger.

### Decomposition of EPSCs

The time of occurrence and the amplitude of individual release events were determined based on deconvolution analysis, as described previously (*Malagon et al., 2016*). In brief, for each synapse, an mEPSC (miniature excitatory postsynaptic current) template was obtained and fitted with a triple-exponential function with five free parameters (rise time, peak amplitude, fast decay time constant, slow decay time constant, and amplitude fraction of slow decay). These five parameters were then used for deconvolution of the template and of individual data traces, producing a narrow spike (called spike template) and sequences of spikes, respectively. Next, each deconvolved trace was fitted with a sum of scaled versions of the spike template, yielding the timing and amplitude of each release event. The amplitude was further corrected for receptor saturation and desensitization, using the exponential relationship between individual amplitudes and the time interval since the preceding release events. Events that were at least 1.7 times larger than the average mEPSC were split into two (*Malagon et al., 2016*) (see below). $s_1$ was determined as the number of SVs released within 5 ms after the first AP of a train. $s_2$ was the number of SVs released within 5 ms after the second AP.

The mean percentage of splitted events was 4.3% in control experiments, 2.9% in 4-AP, and 3.7% after PTP. In response to the first AP, the percentage of splitting was 10.0% in control experiments, 10.3% in 4-AP, and 12.3% after PTP. These results indicate little SV undercounting linked to insufficient time separation between events.

The amplitudes and kinetics of the mEPSC template were unchanged after addition of 4-AP or after PTP. Therefore, the initial template obtained in 3 mM Ca$_o$ was used throughout each experiment. These results confirm that the changes associated with our experimental manipulations are purely presynaptic.

## Changes in latency distributions as a function of *i* and of experimental conditions

As in our previous work, we found that latency distributions were biphasic, and that they could be fitted with a biexponential decay curve. The fast decaying component, with time constant $\tau_f$, reflected the release of docked SVs, while the slow decaying component, with time constant $\tau_s$, reflected two-step release (originating from the RS: *Miki et al., 2018*). Values of $\tau_f$ and $\tau_s$ in 3 mM Ca$_o$ were 0.47 and 4.0 ms, close to previous values in the same condition (*Miki et al., 2018*). The percentage of slow decay grew as a function of *i* from 0% for *i* = 1 to 50% for *i* = 8, again in agreement with our previous study. In 4-AP, $\tau_f$ increased to 0.70 ms, probably reflecting a prolongation of the duration of presynaptic calcium entry, while $\tau_s$ did not change significantly.

When comparing latency distributions before and after addition of 4-AP, we also found a shift of the rising phase of the latency histograms to the right, corresponding to an additional latency component of 0.2 ms. This additional latency likely corresponds to a delay of calcium entry following a lengthening of the AP waveform in the presence of 4-AP.

## *N* determination: variance–mean and binomial analysis

Variance–mean analysis (*Figure 3A, left, B*) was performed using Igor Pro as described previously (*Miki et al., 2016*) to estimate the number of docking sites (*N*). Briefly, plots of var($s_i$) as a function of $<s_i>$ were fitted using a parabola with an initial slope of 1. The intersection of this parabola with the x-axis of the plot gave the maximum *N* value, which when approached to the closest integral, corresponds to the number of docking sites. To obtain $P_i$, $s_i$ is divided by *N*. In conditions of high $p_r$, as in the example in *Figure 3A*, right, the data points approach the intersection of the parabola, as they reach the maximal output of the synapse. The group analysis presented in *Figure 3B* was performed after normalizing mean and variance of individual synapses to *N* = 4.

According to the binomial distribution, the probability that $s_i$, the number of SVs released after AP number *i*, equals *k* is $p(k) = P_i^k (1 - P_i)^{N-k} N!/(k!(N - k)!)$, where $P_i$ is the release probability per docking site and *N* is the number of docking sites (*Malagon et al., 2016*). The values of *N* and *P* were obtained by minimizing the summed squared deviations between the binomial prediction and data (*Figure 3C, D*). This fitting was carried out for $s_1$ and $s_2$ of each synapse and its values compared with the result of the variance–mean curves. This analysis is more robust under high *P* conditions, so the 4-AP data were used to determine *N*. Group average analysis across experimental conditions was performed on synapses with the same number of docking sites (*N* = 4; *Figure 3D*).

## Covariance analysis

Covariance analysis reveals whether the fluctuations of one paramater around its mean value are linked together with the fluctuations of another parameter, either positively (positive covariance) or negatively (negative covariance). In the present work, we analyze the covariance between the sum $S_i = (s_1 + s_2 + \ldots + s_i)$ and $s_{i+1}$. A negative covariance is expected if SVs compete for binding to docking sites, with an amplitude depending on the mode of occupancy of docking sites (*Miki et al., 2016*; *Scheuss and Neher, 2001*). The covariance was determined experimentally as cov($S_i$, $s_{i+1}$) = <$S_i$ − <$S_i$>> <$s_{i+1}$ − <$s_{i+1}$>> and was used to assess the validity of docking site models using Monte Carlo simulations.

## Replenishment in Figure 4

Here, we assess the contribution of replenishment from upstream pools (s transition) to the sum ($P_1 + P_2$) using the model of *Figure 6*. Because this model assumes an initial $\rho$ value of 1, replenishment SVs (originating in the IP) cannot advance beyond the RS during the first inter-AP interval. Consequently, there is no contribution of upstream pools to $P_1$, and the only contribution of upstream pools to $P_2$ comes from RS, following partial replenishment of these sites during the first inter-AP interval. Using the model of *Figure 6* and Monte Carlo simulations, we found that in PTP, the value of $\rho$ at the end of the first inter-AP interval was $\rho_2^{total} = 0.460$, of which the contribution coming from replenishment was $\rho_2^{replenishment} = 0.097$. Because these SVs are in replacement sites, they can contribute to $P_2$ release only in the form of two-step release (*Miki et al., 2018*). From an analysis of the time course of averaged release rate across experiments (as in *Miki et al., 2018*), we estimated the time course of two-step release as an exponential with a time constant of 4 ms, and the amplitude of two-step release at

0% and 20% of total synchronous release after AP number 1 and AP number 2, respectively. For AP number 2, the share of total synchronous release due to two-step release in our 5 ms long measurement window is 2-stepratio = $0.2 \times (1 - \exp(-5/4)) = 0.14$. Combining this ratio with $\rho_2^{replenishment}/\rho_2^{total}$, we obtain the percentage of the $P_2$ response coming from replenishment:

$$P_2^{replenishment}/P_2^{total} = (\rho_2^{replenishment}/\rho_2^{total}) \times (2 - stepratio) = (0.097/0.460) \times 0.14 = 0.03$$

The corresponding ratio for $P_1$ is $P_1^{replenishment}/P_1^{total} = 0$ since 2-stepratio = 0 after the first AP.

Given the experimental values $P_1^{total} = 0.88$ and $P_2^{total} = 0.67$ (*Figure 4*), it can be calculated that 1.3% of the sum ($P_1 + P_2$) comes from replenishment SVs in PTP. Lower percentage values would apply in control or in 4-AP, which have lower 2-stepratio values than PTP. Overall, these calculations suggest that the contribution of replenishment SVs to the sum ($P_1 + P_2$) may be neglected.

## Model and simulation of SV docking and release

Monte Carlo simulation of two-step SV docking and release (*Figure 6*) was performed using Igor Pro as described previously (*Miki et al., 2016*; *Tran et al., 2022*). For *Figure 6—figure supplement 1*, the IP was added to the RS/DS model and transitions and replenishment conditions were adapted accordingly.

In the LS/TS model transitions are equivalent to the RS/DS model (*Figure 6—figure supplement 2*), however only one SV can occupy the DS at a time, being either in TS or LS. The replenishment step was adapted to fit this condition. As in the RS/DS model, the distal–proximal (TS–LS) transition is sequential and release can only happen from distally bound (TS) vesicles.

For the models, parameter values are shown in figures or figure legends and were obtained by fitting using least mean squares of free parameters. For the RS/DS model with IP, parameters were kept as in the RS/DS original fitting and a new fitting was performed for the two new parameters (IP size and IP replenishment probability). Calculations were done 5000 times for each condition and the averaged values are displayed.

## Acknowledgements

We thank Profs. Erwin Neher and Takafumi Miki for helpful comments on the manuscript. This work was supported by CNRS (UMR 8003), by Fondation pour la Recherche Médicale (grant SPF201809007190 to VT) and by the European Community (ERC Advanced Grant 'Single Site' nb. 294509 to AM).

## Additional information

### Funding

| Funder | Grant reference number | Author |
| --- | --- | --- |
| Fondation pour la Recherche Médicale | SPF201809007190 | Van Tran |
| European Research Council | 294509 | Alain Marty |

The funders had no role in study design, data collection, and interpretation, or the decision to submit the work for publication.

### Author contributions

Melissa Silva, Data curation, Investigation, Writing - review and editing; Van Tran, Conceptualization, Data curation, Formal analysis, Supervision, Methodology, Writing - review and editing; Alain Marty, Conceptualization, Supervision, Funding acquisition, Writing - original draft, Project administration, Writing - review and editing

### Author ORCIDs

Melissa Silva http://orcid.org/0000-0002-2687-8650
Van Tran http://orcid.org/0000-0002-3435-0258
Alain Marty http://orcid.org/0000-0001-6478-6880

### Ethics

C57BL/6 mice (10–16 days old; either sex) were used for preparation of acute brain slices. Animals were housed and cared for in accordance with guidelines of Université Paris Cité (approval no. D 75-06-07).

Reviewer #1 (Public Review): https://doi.org/10.7554/eLife.91087.3.sa1
Reviewer #2 (Public Review): https://doi.org/10.7554/eLife.91087.3.sa2
Author Response https://doi.org/10.7554/eLife.91087.3.sa3

## Additional files

### Supplementary files

• MDAR checklist

### Data availability

The data that support the findings of this study are openly available in *Figure 2—source data 1* (individual experimental data for *Figures 2, 4–6*), and in Dryad at https://doi.org/10.5061/dryad.9cnp5hqr8 (individual experimental data for *Figure 3*).

The following dataset was generated:

| Author(s) | Year | Dataset title | Dataset URL | Database and Identifier |
|---|---|---|---|---|
| Silva M, Tran V, Marty A | 2023 | Data from: A maximum of two readily releasable vesicles per docking site at a cerebellar single active zone synapse | https://doi.org/10.5061/dryad.9cnp5hqr8 | Dryad Digital Repository, 10.5061/dryad.9cnp5hqr8 |

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
