## [Editor Report · eLife assessment]

The study used slice physiology and modeling to investigate neurotransmitter release at the cerebellar parallel fiber-to-molecular layer interneuron synapse, revealing that each docking site can accommodate up to two synaptic vesicles simultaneously. The evidence presented is **convincing**. These **important** findings validate a two-step docking model and shed light on the mechanisms underlying short-term synaptic plasticity and strategies for achieving synaptic reliability, which plays a critical role in information processing in the brain.

---

## [Referee Report · Reviewer #1 (Public Review)]

Summary: By elevating Ca influx and inducing PTP, the authors have maximised the release probability. In this condition, the release probability is nearly one. Under such a condition, the release site can release another vesicle in a short time. By analyzing mean, variance and covariance, the authors propose a release model that each release site contains a docking site and a replacement site. They excluded the LS-TS model (Neher and Brose) based on discrepancy between model and the data (mean and covariance).

Strengths: The authors have used a minimal stimulation and modelling nicely to look into stochastic nature of release sites with good resolution. This cannot be done at other synapses. Overall conclusions are reasonable and convincing.

Weaknesses: The interpretation is somewhat model-dependent, and it is unclear if the interpretation is unique. For example, it is unclear if the heterogeneous release probability among sites, silent sites, can explain the results. However, the authors discuss these potential caveats in a fair manner and argue that their model is very likely to be the best so far.

---

## [Referee Report · Reviewer #2 (Public Review)]

Summary:

Silva et al. describe an experimental study conducted on cerebellar parallel fiber-to-molecular interneuron synapses to investigate the size of the readily releasable pool (RRP) of synaptic vesicles (SVs) per docking site in response to trains of action potentials. The study aims to determine whether there are multiple binding sites for SVs at each docking site, which could lead to a higher RRP size than previously thought.

The researchers used this glutamatergic synapse to conduct their experiments. They employed various techniques and manipulations to enhance release probability, docking site occupancy, and synaptic depression. By counting the number of released SVs in response to action potential trains and normalizing the results based on the number of docking sites, they estimated the RRP size per docking site.

The key findings and observations in the manuscript are as follows:

Docking Site Occupancy and Release Probability Enhancement: The researchers used 4-amidopyridine (4-AP) and post-tetanic potentiation (PTP) protocols to enhance the release probability of docked SVs and the occupancy of docking sites, respectively.

Synchronous and Asynchronous Release: Synchronous release refers to SVs released in response to individual action potentials, while asynchronous release involves SVs released after the initial release response due to calcium elevation. The study observed changes in the balance between synchronous and asynchronous release under different conditions, revealing the degree of filling of the RRP.

Modeling of Release Dynamics: The researchers employed a modeling approach based on the "replacement site/docking site" (RS/DS) model, where SVs bind to a replacement site before moving to a docking site and eventually undergoing release. The model was adjusted to experimental conditions to estimate parameters like docking site occupancy and release probabilities.

Comparison of Different Models: The study compared the RS/DS model with an alternative model known as the "loosely docked/tightly docked" (LS/TS) model. The LS/TS model assumes that a docking site can only accommodate one SV at a time, while the RS/DS model considers the possibility of accommodating multiple SVs.

Maximum RRP Size: Through a combination of experimental results and model simulations, the study revealed that the maximum RRP size per docking site reached close to two SVs under certain conditions, supporting the idea that each docking site can accommodate multiple SVs.

Strengths:

The study is rigorously conducted and takes into consideration previous work of RRP size and SV docking site estimation. The study addresses a long-standing question in synaptic physiology.

Weaknesses:

It remains unclear how generalizable the findings are to other types of synapses.

---

## [Author Response]

The following is the authors’ response to the original reviews.

**Reviewer 1 (Public review):**
Weaknesses: The interpretation is somewhat model-dependent, and it is unclear if the interpretation is unique. For example, it is unclear if the heterogeneous release probability among sites, silent sites, can explain the results. N estimates out of variance-mean analysis for example may be limited by the availability of postsynaptic receptors.

To address this criticism, we have added a paragraph in the Discussion outlining the main assumptions underlying our work and how possible deviations from these assumptions may have affected our conclusions. This new paragraph is titled ' Assumptions behind our analysis, and possible limitations of our conclusions'.

**Reviewer 1, Recommendations to Authors:**
Without molecular evidence or anatomical evidence, the model and conclusions may remain as a postulate at this stage. This can be discussed carefully. Also, the study looks a bit narrow regarding the scope, only dealing with RS-DS model vs TS-LS model. Maybe, the authors pick up a bit more qualitative findings that directly support RS-DS model.

To address these issues, another paragraph has been added to the Discussion titled 'Functional evidence in favor of the RS/DS model at PF-MLI synapses, and remaining uncertainties on the molecular composition and morphological arrangement of docking sites'.

Minor: Fukaya et al. studied not cerebellar mossy fiber synapses.

We apologize for this error, which has now been rectified.

**Reviewer 2 (Public review):**
It remains unclear how generalizable the findings are to other types of synapses.

We agree with the Reviewer: this is a limitation of our study. In the Discussion we have a paragraph titled 'Maximum RRP size for other synaptic types' where we discuss this point. As we say in this paragraph, central synapses are clearly diverse, and the level of applicability of our results across preparations will depend on our ability to extend SV counting to various types of brain synapses. For the moment SV counting has been applied to only two types of synapses: PF-MLI synapses and hMF-IN synapses. We are encouraged by the fact that the simple synapse study by Tanaka et al. (2021), carried out at hMF-IN synapses, offers another example where the ratio between RRP size and N is larger than 1.

Recommendations to Authors,Minor comments:The manuscript is at times difficult to read or reads like a review. The introduction could be shortened to concisely outline the motivation and premises for the study. The results and methods sections should not contain excessive interpretation and discussion. Although very informative, it distracts from the simple principal message.

To address these criticisms, we have shortened the Introduction and parts of the Results section. These changes have resulted in a presentation of Results that is shorter and more focused on data and simulations than in the previous version. Nevertheless, readers need to be informed of ongoing research on docking sites and the principles of sequential models to understand the usefulness of our work. For this reason, we have maintained a theoretical section at the beginning of Results.

The rationale for the choice of synapse and experimental conditions remains unclear until the discussion. This needs to be clearly addressed at the beginning, in the introduction, or in the results. In particular, the extracellular calcium concentration and the addition of 4-AP to the recording solution should be addressed in the results.

The reason to choose the PF-MLI synapse is now indicated at the end of the Introduction. The rationale underlying our choice of experimental conditions including the extracellular calcium concentration and the addition of 4-AP is now briefly explained in the beginning the second section of Results (titled 'Maximizing RRP size and its release during AP trains'), and more extensively in the Methods section (as in the previous version of the manuscript).

Potential confounds of the approach should be discussed e.g. could a broadened AP in 4-AP alter synchronicity of release, i.e. desynchronization of release, especially during trains. That could be complemented with information on the EPSC kinetics (rise, decay) under different experimental conditions, as well as during train stimulation. How could presynaptic calcium concentration and time course in 4-AP impact the conclusions?

To study the effects of 4-AP on AP broadening we have performed a new analysis of EPSC latencies in control and in 4-AP. In both cases the first latencies were independent of i. In 4-AP, first latencies displayed a small right shift of 0.2 ms (see additional figure below). This indicates that 4-AP does broaden the AP waveform, but that the extent of this broadening is limited. This new information has been added in the Methods of the revised manuscript.

As suspected by the Reviewer, the latency distribution changes as a function of i and in the presence of 4-AP. Consistent with earlier findings (Miki et al., 2018), the proportion of 2-step release (with longer latencies) augments as a function of i both in control and in 4-AP. We also find that the value of the fast time constant of the latency distribution,τf, is larger in 4-AP than in control. This last result probably indicates a longer presynaptic calcium entry in 4-AP.

In the revised version, we describe these results in the Methods section, in a new paragraph titled 'Changes in latency distributions as a function of i and of experimental conditions'.

While the latency distributions change as a function of i and as a function of experimental conditions, this does not affect our conclusions, because these conclusions are based on the summed number of release events after each AP (or in other words, on the integral of the latency distributions).

The kinetics of mEPSCs (risetime and decay time) are unchanged by 4-AP or by PTP. Consequently, in a given experiment, we used the same template to perform our deconvolution analysis for all conditions that were examined (starting with 3 mM Cao up to 200 Hz). This information has now been added in Methods.

Following an AP stimulation, the amount of calcium entry in the presence of 4-AP is presumably much larger than in control. TEA, a weaker K channel blocker than 4-AP at PF-MLI synapses, elicits a marked increase in calcium entry (Malagon et al., 2020). This suggests an even larger increase with 4-AP, even though this has not been directly confirmed in the present work. The enhanced calcium entry translates in an increase in the parameters pr, r and s of our model. The important thing for our study is to increase pr and r as much as possible to promote the emptying of the RRP during trains. Knowing the exact amount of calcium entry and its relation to pr /r increase is not essential for this purpose. Likewise, whether r (and/or s) increase as a function of i is of little practical importance since much of the RRP is emptied already after the second stimulation, at least in the most extreme case (200 Hz stimulation).

The applicability of this model to other synapses needs to be addressed more thoroughly. This synapse, under physiological conditions, has a very low Pr, and the experimental conditions have to be adjusted dramatically to achieve a high-Pr. How applicable are the conclusions to high-Pr synapses and/or synapses that operate in a multivesicular release regime? Although that might be difficult to test experimentally it should be addressed in the discussion.

The applicability issue to other synapses has been addressed above, in response to the public comments of the same Reviewer.

As the Reviewer points out, the PF-MLI synapse has a small P value under physiological conditions. One can speculate that synapses that exhibit a higher P value may have a higher docking site occupancy than PF-MLI synapses. This feature would increase their chance of having a ratio of RRP size over N larger than 1, as it occurs in PF-MLI synapses in high docking occupancy conditions. A sentence making this point has been added to the paragraph titled 'Maximum RRP size for other synaptic types' in the revised manuscript.

**Author response image 1. sa3fig1:** Latency histograms for s1 in control and in the presence of 4-AP. After normalization, the averaged latency histogram in 4-AP displays an additional delay of 0.2 ms, and a slowing of the time constant τf from 0.47 ms to 0.70 ms.